# ANY-SHOT, ANY-WAY META-LEARNING BY BAYESIAN NONPARAMETRIC DEEP EMBEDDING

## ABSTRACT

Learning at small or large scales of data is addressed by two strong but divided frontiers: few-shot learning and standard supervised learning. Few-shot learning focuses on sample efficiency at small scale, while supervised learning focuses on accuracy at large scale. Ideally they could be reconciled for effective learning at any number of data points (shot) and number of classes (way). To span the full spectrum of shot and way, we frame the *variadic learning* regime of learning from any number of inputs. We approach variadic learning by meta-learning a novel multi-modal clustering model that connects bayesian nonparametrics and deep metric learning. Our bayesian nonparametric deep embedding (BANDE) method is optimized end-to-end with a single objective, and adaptively adjusts capacity to learn from variable amounts of supervision. We show that multi-modality is critical for learning complex classes such as Omniglot alphabets and carrying out unsupervised clustering. We explore variadic learning by measuring generalization across shot and way between meta-train and meta-test, show the first results for scaling from few-way, few-shot tasks to 1692-way Omniglot classification and 5k-shot CIFAR-10 classification, and find that nonparametric methods generalize better than parametric methods. On the standard few-shot learning benchmarks of Omniglot and mini-ImageNet, BANDE equals or improves on the state-of-the-art for semi-supervised classification.

## 1 INTRODUCTION

In machine learning, classification problems span two important axes: the number of classes to recognize (the "way" of the problem) and the number of examples provided for each class (the "shots" to learn from). At one extreme, there are large-scale tasks like ImageNet in which there are 1000 classes with roughly 1000 examples each (a 1000-way, $\sim$1000-shot problem). At the other extreme, there are datasets for learning from few examples, such as Omniglot, which features a 5- or 20-way, 1-shot problem. State-of-the-art methods for these two learning regimes are substantially different, with the former dominated by standard parametric deep networks and the latter by episodic meta-learning techniques. Moreover, as shown in our experiments, many methods degrade when the shot and way vary between training and testing. By contrast, humans recognize both familiar and unfamiliar categories whatever the amount of data, and can even learn a new category from a single example (Lake et al., 2015). To this end, we introduce a learning problem which requires generalization from few-way, few-shot problems to many-way, many-shot problems.

We call this regime of variable shot and way the *variadic learning* regime, after variadic functions. Just as variadic functions are those which can take any number of arguments to produce a result, a good variadic learner must learn from any amount of data, whatever the number of examples and classes, and produce strong results across unknown data distributions during test.

Meta-learning provides one potential avenue for pursuing a variadic learner. Meta-learning approaches generally use plentiful supervision from one distribution of tasks to learn an algorithm or metric that can be applied to more sparsely supervised tasks. Ideally, meta-learning approaches do not need knowledge of the specific setting in which they will be used. However, in practice, meta-learning approaches have commonly been trained and evaluated in constrained circumstances, so their generalization properties are not fully known. Perhaps most significantly, meta-learning is usually carried out independently across settings so that a different learner is specialized to each

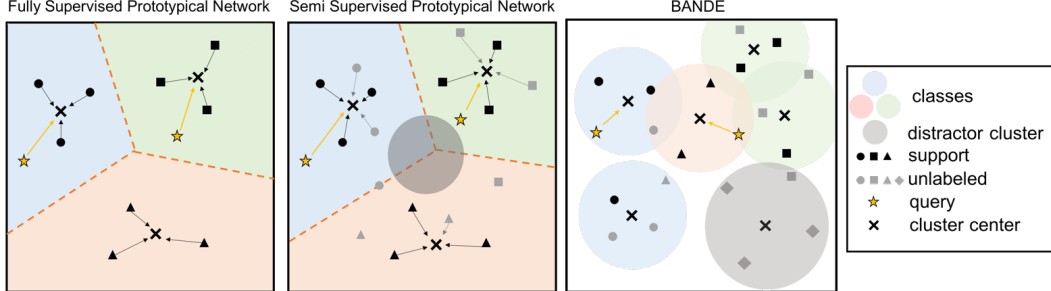

Figure 1: Our bayesian bonparametric deep embedding (BANDE) method is optimized end-to-end to cluster labeled and unlabeled data into *multi-modal, many-to-one* prototypes. BANDE represents each class by a set of clusters, unlike prior prototypical methods that are limited to uni-modal, one-to-one representation of each class by a single cluster. Multi-modal prototypes make BANDE more accurate on complex classes like alphabets and more general for fully-supervised, semi-supervised, and even unsupervised clustering where prior methods are undefined.

$n$-way, $k$-shot task. This potentially limits their deployment to more diverse settings with variable shot and way that we address in this work.

As a first step towards a strong variadic learner, we propose a multi-modal (many-to-one) semi-supervised clustering approach which can adapt its capacity to the underlying class representations, and show that this is critical for modeling more complex data distributions. This innovation allows our model to perform inference with any amount of supervision (from totally unsupervised to fully supervised) after training, and adjust better to variable shot and way than existing approaches.

Our bayesian nonparametric deep embedding (BANDE) model (see Figure 1) extends prototypical networks to multi-modal clustering. Clustering with multiple modes is critical for complex classes, and multi-modality makes unsupervised clustering possible. BANDE generalizes across any-shot, any-way tasks better than existing methods. At the many-way extreme, when trained with 5-way 1-shot episodes, BANDE achieves $75\%$ accuracy for 1692-way 10-shot classification of Omniglot, improving on both few-shot and supervised learning baselines. At the many-shot extreme, BANDE approaches the accuracy of a standard supervised learner on CIFAR-10/100. On standard few-shot benchmarks BANDE is state-of-the-art in the semi-supervised setting.

## 2 RELATED WORK

**Prototypes and Nonparametrics** Prototypical networks (Snell et al., 2017) and semi-supervised prototypical networks (Ren et al., 2018) are the most closely related to our work. Prototypical networks simply and efficiently represent each class by its mean in a learned embedding. They assume that the data is fully labeled. Ren et al. (2018) extend prototypes to the semi-supervised setting by refining prototypes through soft k-means clustering of the unlabeled data. They assume that the data is at least partially labeled. Snell et al. (2017) and Ren et al. (2018) are limited to one cluster per class. We define a more general and adaptive approach through bayesian nonparametrics that extends prototypical networks to multi-modal clustering, with one or many clusters per class, of labeled and unlabeled data alike. Through multi-modal representation and adaptive inference of the number of modes, our method is significantly more accurate on complex classes, does unsupervised clustering, and improves on standard semi-supervised few-shot learning benchmarks.

For multi-modal clustering we incorporate DP-means (Kulis & Jordan, 2012) in our method. DP-means is a scalable, bayesian nonparametric algorithm for unsupervised clustering that creates new clusters when data points are more than a threshold $\lambda$ away from existing clusters. Our full method handles labeled and unlabeled data, augments the clustering with soft assignments under a normalized Gaussian likelihood, and defines a procedure to choose $\lambda$ during learning and inference.

**Metric Learning** Learning a metric to measure a given notion of distance/similarity addresses recognition by retrieval: given an unlabeled example, find the closest labeled example. The contrastive loss and siamese network architecture (Chopra et al., 2005; Hadsell et al., 2006) optimize an embedding for metric learning by pushing similar pairs together and pulling dissimilar pairs apart. Of particular note is research in face recognition, where a same/different retrieval metric is used for many-way classification (Schroff et al., 2015). Our approach is more aligned with metric learning by

meta-learning (Koch, 2015; Vinyals et al., 2016; Snell et al., 2017; Garcia & Bruna, 2018). These approaches meta-learn a distance function by directly optimizing the task loss, such as cross-entropy for classification, through episodic optimization (Vinyals et al., 2016) for each setting of way and shot. While we likewise learn by episodic optimization, we differ from previous meta-learning work in our examination of generalization to variable numbers of examples and classes during testing, and show improvement in this regime. Unlike metric learning on either exemplars (Schroff et al., 2015) or prototypes (Snell et al., 2017; Ren et al., 2018), our method adaptively interpolates between exemplar and uni-modal prototype representation by deciding the number of modes during clustering.

**Learning Regimes** Variadic learning is best explained in relation to few-shot learning, low-shot learning, and conventional supervised learning. Few-shot learning (Fei-Fei et al., 2006; Vinyals et al., 2016) handles tasks of fixed, known, and small numbers of data points and classes. In contrast, variadic tasks have variable numbers of data points and classes that can shift across tasks. Low-shot learning (Hariharan & Girshick, 2017; Qi et al., 2018; Qiao et al., 2018) addresses both densely supervised base classes and sparsely supervised novel classes, but presupposes which classes are in which set. Variadic learning also addresses these extremes of supervision, but requires no knowledge of how much or how little supervision each class has. Large-scale supervised learning (Bottou, 2010) parameterizes the model by the number of classes, and is tuned to the amount of data by choosing capacity, optimization schedules, and so forth. Variadic learning requires accuracy without specialization to shot and way. Life-long learning (Thrun, 1996; 1998) concerns variable shot and way for streams of non-stationary problems, while variadic learning is for one problem of unknown dimensions. Bridging life-long and variadic learning is sensible but out of scope for this work.

## 3 BAYESIAN NONPARAMETRIC DEEP EMBEDDINGS (BANDE)

Our method end-to-end learns a deep embedding network and jointly clusters labeled and unlabeled data points by bayesian nonparametrics. Crucially, our model is able to express a single class as multiple modes, unlike the uni-modal clustering approaches of prior work. Figure 1 gives a schematic view of our multi-modal representation and how it differs from prior prototypical representations. Algorithm 1 expresses one step of model optimization in pseudocode.

**Few-shot Meta-learning** In few-shot classification we are given a *support* set $S = \{(x_1, y_1), \ldots, (x_K, y_K)\}$ of $K$ labeled examples and a *query* set $Q = \{(x'_1, y'_1), \ldots, (x'_{K'}, y'_{K'})\}$ of $K'$ labeled examples where each $x_i, x'_i \in \mathbb{R}^D$ is a $D$-dimensional feature vector and $y_i, y'_i \in \{1, \ldots, N\}$ is the corresponding label. In the semi-supervised setting, $y_i$ may not be provided for every example $x_i$. The support set is for learning while the query set is for inference: the few-shot classification problem is to recognize the class of the queries given the labeled supports.

Meta-learning is carried out by episodic optimization of the model parameters for the task loss. *Episodes* are comprised of support and query sets, constructed by randomly sampling a subset of classes, sampling examples from these classes, and then partitioning the examples into supports and queries. Optimization iterates by making one episode and one update. The update is defined by the task loss, which for classification could be the softmax cross-entropy loss.

For deep metric learning models like ours, the model parameters are those of the embedding function $h_\phi : \mathbb{R}^D \to \mathbb{R}^M$ that is a deep network with parameters $\phi$. The embedding of an example $x$ is the $M$-dimensional feature vector taken from the last layer of the network.

Meta-training proceeds by optimizing the model parameters $\phi$ with respect to a task loss. Meta-testing proceeds episodically like meta-training but without query labels or further optimization.

**Prototypes** Prototypical networks (Snell et al., 2017) take the mean of the embedded support examples of a particular class to form a *prototype*: $\mu_n = \frac{1}{|S_n|} \sum_{(x_i, y_i) \in S_n} h_\phi(x_i)$, with $S_n$ denoting the set of support examples of class $n$. In conjunction with a distance function $d(x_i, x_j)$, this provides an inference scheme for a query point $x$ as the softmax over distances to the prototypes: $p_\phi(y = n \mid x) = \frac{\exp(-d(h_\phi(x), \mu_n))}{\sum_{n'} \exp(-d(h_\phi(x), \mu_{n'}))}$. $\phi$ is optimized by minimizing the negative log-probability of the true class of each query point by stochastic gradient descent in each episode. Prototypical networks defined in this way learn to create *uni-modal* class distributions for *fully labeled* supports.

**Multi-modal Clustering** Our method defines *multi-modal* prototypes of both labeled and unlabeled data. That is, a single class is represented by a *set* of cluster modes. By deciding the number of modes, our method interpolates between exemplar and uni-modal prototype representations, in effect adjusting its capacity depending on the data.

To create multi-modal prototypes, we extend the non-parametric clustering algorithm DP-means (Kulis & Jordan, 2012) to make it compatible with end-to-end learning. DP-means iterates through all examples in a dataset, computing the example's minimum distance to all existing cluster means. If this distance is greater than a particular threshold $\lambda$, a new cluster is created with mean $h_\phi(x_i)$, the example assigned to it. If $x_i$ is labeled, the new cluster takes on its label. While we use DP-means for cluster creation, we include cluster variances for reassignment. Labeled clusters are assigned a variance $\sigma_l$ and unlabeled clusters are assigned a variance $\sigma_u$. $\sigma_l$ and $\sigma_u$ are differentiable, and therefore learned along with the embedding parameters $\phi$. (We discuss the probabilistic interpretations of this choice in the next section.)

$\lambda$, the threshold for creating a new cluster, is the sole hyperparameter for DP-means clustering. It is non-differentiable, and so it cannot be learned jointly. Instead, we set $\lambda$ episodically as a function of the data. In Kulis & Jordan (2012), $\lambda$ is parameterized as $-2\sigma \log(\frac{\alpha}{(1+\frac{\rho}{\sigma})^{d/2}})$. $\alpha$ is the relative probability of forming a new cluster in the Chinese Restaurant Process prior (Aldous, 1985), and $\rho$ is a measure of the standard deviation for the base distribution from which clusters are assumed to be drawn. We estimate $\rho$ as the variance in the labeled cluster means within an episode, while $\alpha$ is treated as a hyperparameter. In our experiments, we found a wide range of $\alpha$ values to give similar results, with the embeddings adjusting their overall magnitudes to match the magnitude of $\alpha$.

## 3.1 PROBABILISTIC INTERPRETATIONS OF HARD AND SOFT CLUSTERING

The choice of hard or soft clustering has theoretical ramifications. There are three clustering variants to consider: fully hard, fully soft, and hybrid hard-soft.

Fully hard clustering corresponds to following DP-means in a theoretically exact manner, with both $\sigma_u$ and $\sigma_l$ set to 0, and the UPDATEASSIGNMENTS function assigning $z_i = argmin_c[d_{i,c}]$ for each example $i$. This variant is theoretically precise as an extension of DP-means for end-to-end learning and simultaneous clustering of labeled and unlabeled data. Fully soft clustering corresponds to an extension and reinterpretation of prior work on semi-supervised prototypical networks (Ren et al., 2018) (see Section A.4). Through the lens of bayesian nonparametrics, we derive this connection to an approximation of the Chinese Restaurant Process (CRP) (Aldous, 1985) in Section A.4 of the appendix. While fully hard and fully soft clustering admit clearer probabilistic interpretations, they are empirically less accurate than hybrid hard-soft clustering. Table 1 compares the variants on a standard semi-supervised few-shot learning benchmark (detailed further in Section 4.3).

BANDE does hard-soft clustering throughout our experiments. For hard-soft clustering, UPDATEASSIGNMENTS does soft assignment of $z_{i,c} = \frac{\mathcal{N}(h_\phi(x_i);\mu_c,\sigma_c)}{\sum_c \mathcal{N}(h_\phi(x_i);\mu_c,\sigma_c)}$ for all examples $i$.

Table 1: Clustering comparison on 5-way 1-shot semi-sup. Omniglot.

| Clustering | Accuracy |
|---|---|
| Hard-Hard | 97.0 |
| Soft-Soft | 98.4 |
| BANDE (Hard-Soft) | 99.0 |

## 3.2 CUMULATIVE SUPERVISION

We extend BANDE into a cumulative variant, BANDE-C, that accumulates supervision non-episodically by remembering prototypes across episodes. Concretely, we initialize the cluster means $\mu_c$ by including a cluster mean from memory, $\phi_{m,c}$, with the current episodic sample mean (i.e. $\mu_c = \frac{1}{|z_i \in C|+1}(\phi_{m,c} + \sum_{i,z_i \in c} h_\phi(x_i))$. $\phi_{m,c}$ is computed as if $c$ was uni-modal, regardless of whether the clustering was multi-modal in a previous episode. Since the embedding representation rapidly changes early in training, we introduce a discount factor on the stored embedding $\gamma\phi_{m,c}$ proportional to the current learning rate. Whenever the class is encountered in a future episode, we update the remembered prototype with the cluster mean after episodic inference. We only experiment with BANDE-C in the variadic setting (Section 4.2); everywhere else we keep standard episodic training and testing.

Note that standard prototypical networks can likewise be augmented to remember prototypes and non-episodically accumulate supervision in this manner.

---

**Algorithm 1** BANDE: one optimization episode. $n_s$ is the number of labeled classes (way) and $k_s$ is the number of labeled examples of each class (shot). $k_q$ is the number of query examples per class. For a set $A$, $A_n$ is the subset of $A$ with all examples of class $n$. $p(x|\mu, \sigma)$ is the Gaussian density.

---

**Input:** support set $S$, query set $Q$, and unlabeled set $U$.
**Output:** loss $J$ for the episode.

$\quad C \leftarrow n_s$ $\qquad\qquad\qquad\qquad\qquad\qquad\qquad\qquad\qquad$ $\triangleright$ $C$ is the total number of clusters
$\quad$ **for** $c \in \{1, ..., C\}$ **do**
$\quad\quad l_c \leftarrow c$ $\qquad\qquad\qquad\qquad\qquad\qquad\qquad\qquad$ $\triangleright$ $l_c$ is the cluster label
$\quad\quad \mu_c \leftarrow \frac{1}{k_s} \sum_{(x_i, y_i) \in S_c} h_\phi(x_i)$ $\qquad\qquad\qquad\qquad$ $\triangleright$ $\mu_c$ is the cluster mean
$\quad\quad \sigma_c \leftarrow \sigma_l$ $\qquad\qquad\qquad\qquad\qquad\qquad\qquad$ $\triangleright$ $\sigma_c$ is the cluster variance
$\quad$ **end for**
$\quad$ $\triangleright$ Iterate over the labeled and unlabeled data and create new clusters.
$\quad$ **for** each example $i \in S \cup U$ **do**
$\quad\quad$ **for** $c$ in $\{1, ..., C\}$ **do**
$\quad\quad\quad d_{i,c} \leftarrow \begin{cases} \|h_\phi(x_i) - \mu_c\|^2 \text{ if (example } i \text{ is labeled and } l_c = y_i) \text{ or example } i \text{ is unlabeled} \\ +\infty \text{ otherwise} \end{cases}$
$\quad\quad$ **end for**
$\quad\quad$ **if** $\min_c(d_{i,c}) > \lambda$ **then**
$\quad\quad\quad C \leftarrow C + 1$
$\quad\quad\quad l_C \leftarrow y_i$ $\qquad\qquad\qquad\qquad\qquad$ $\triangleright$ Cluster takes the label of the example
$\quad\quad\quad \mu_C \leftarrow h_\phi(x_i)$ $\qquad\qquad\qquad$ $\triangleright$ Cluster mean takes the embedding of the example
$\quad\quad\quad \sigma_C \leftarrow \begin{cases} \sigma_l, \text{ if } y_i \neq 0 \\ \sigma_u, \text{ otherwise} \end{cases}$
$\quad\quad$ **end if**
$\quad$ **end for**
$\quad z \leftarrow \text{UPDATEASSIGNMENTS}(\{h_\phi(x)\}, \mu, \sigma)$ $\qquad$ $\triangleright$ Update all cluster-example assignments
$\quad \mu \leftarrow \{\frac{\sum_i z_{i,c} h_\phi(x_i)}{\sum_i z_{i,c}} \mid c \in 1, ..., C\}$ $\qquad\qquad$ $\triangleright$ Update all cluster means
$\quad$ $\triangleright$ Cross-entropy loss on the most probable cluster of the true class and all clusters of other classes
$\quad J \leftarrow 0$
$\quad$ **for** $n$ in $\{1, ..., n_s\}$ **do**
$\quad\quad c^* \leftarrow \underset{c:l_c=n}{\arg\max} \log p(x|\mu_c, \sigma_c)$
$\quad\quad J \leftarrow J + \frac{1}{n_s k_q} \sum_{(x,y) \in Q_n} \left[ -\log p(x|\mu_{c^*}, \sigma_{c^*}) + \log\left( \sum_{c':l_{c'} \neq n} p(x|\mu_{c'}, \sigma_{c'}) + p(x|\mu_{c^*}, \sigma_{c^*}) \right) \right]$
$\quad$ **end for**

---

# 4 EXPERIMENTS

We experimentally show that multi-modal prototypes are more accurate and more general than uni-modal prototypes. In our new variadic setting for any-shot, any-way learning we explore which methods do (and do not) generalize across shot and way. We report the first results for extreme generalization to 1692-way classification and 5000-shot from few-shot episodic optimization. For few-shot learning, we show competitive results for few-shot fully-supervised and semi-supervised classification on the standard benchmarks of Omniglot and mini-ImageNet.

We control for architecture and optimization by comparing methods with the same base architecture and same episodic optimization settings. All code for our method and baselines will be released.

For these experiments we make use of standard few-shot and supervised learning datasets and furthermore define new variadic evaluation protocols on these common benchmarks. We consider Omniglot and mini-ImageNet, two widely used datasets for few-shot learning research, and CIFAR-10/CIFAR-100, two popular datasets for supervised learning research with deep networks.

**Omniglot** (Lake et al., 2015) is a dataset of 1,623 handwritten characters from 50 alphabets. There are 20 examples of each character, where the images are resized to 28x28 pixels and each image is

rotated by multiples of $90°$. This gives 6,492 classes in total, which are then split into 4,112 training classes, 1,692 test classes and 688 validation classes.

**mini-ImageNet** (Vinyals et al., 2016) is a reduced version of the ILSVRC'12 dataset (Russakovsky et al., 2015), which contains 600 84x84 images for 100 classes randomly selected from the full dataset. We use the split from Ravi & Larochelle (2017) with 64/16/20 classes for train/val/test.

**CIFAR-10/100** (Krizhevsky & Hinton, 2009) are classification datasets of 32x32 color images drawn from the Tiny Images project (Torralba et al., 2008). CIFAR-10 has 10 classes and CIFAR-100 has 100 classes (plus 20 super-classes). Both have 50k training images and 10k testing images and both are balanced so that every class has an equal number of images.

## 4.1 Accuracy and Generality of Multi-modal Prototypes

Our experiments on Omniglot alphabets and characters show that multi-modal prototypes are significantly more accurate than uni-modal prototypes for recognizing complicated classes (alphabets) and recover uni-modal prototypes as a special case for recognizing simple classes (characters). Multi-modal prototypes generalize better for super-class to sub-class transfer learning, improving accuracy when meta-training on alphabets but meta-testing on characters. By unifying the clustering of labeled and unlabeled data alike, our multi-modal prototypes even address fully unsupervised clustering, unlike prior prototypical network models that are undefined without labels.

We first show the importance of multi-modality for learning representations of multi-modal classes: Omniglot alphabets. For these experiments we meta-train for alphabet classification, using only the super-class labels. Episodes are constructed by sampling 1 example of 200 different random characters in the support set, with 5 examples of each character in the query.

Table 2: Alphabet and character recognition accuracy. BANDE improves accuracy for multi-modal alphabet classes, preserves accuracy for uni-modal character classes (Chars), and generalizes better from super-classes to sub-classes.

| Training | Testing | Proto. Nets | BANDE |
|----------|---------|-------------|-------|
| Alphabet | Alphabet | 64.9±0.2 | **91.2**±0.1 |
| Alphabet | Chars (20-way) | 85.7±0.2 | **95.3**±0.2 |
| Chars | Chars (20-way) | 94.9±0.2 | **95.1**±0.1 |

For alphabet testing, we provide 100 randomly selected characters with alphabet labels in the support, making this a mixed-shot problem. For character testing, we provide 1 labeled image of 20 different characters as support, and score based on correct character assignments of the queries. As seen in table 2, in both testing configurations, BANDE substantially outperforms prototypical networks.

On 20-way 1-shot character recognition, BANDE achieves $95.3\%$ from alphabet supervision alone, slightly out-performing prototypical networks trained directly on character recognition ($94.9\%$).

**Fully Unsupervised Clustering** BANDE is able to do fully unsupervised clustering during meta-test via multi-modality. Prior work on prototypical networks (Snell et al., 2017) and semi-supervised prototypical networks (Ren et al., 2018) cannot address this setting because the models are undefined without labeled data.

BANDE handles labeled and unlabeled data by the same clustering rule, inferring the number of clusters as needed, and achieves good accuracy under the standard clustering metrics of normalized mutual information (NMI) and purity. We examine BANDE's clustering performance in Table 3 by randomly sampling 5 examples of $n$ classes from the test set and treating them as unlabeled samples. BANDE maintains remarkably strong performance across a large number of unlabeled clusters, without knowing the number of classes in advance, and without having seen any examples from the classes during training.

Table 3: Unsupervised clustering by BANDE. BANDE clusters with high purity and normalized mutual information (NMI) across a wide range of unseen classes, while prior prototypical methods are undefined.

| Metric | 10-way | 100-way | 200-way |
|--------|--------|---------|---------|
| Purity | 0.97 | 0.76 | 0.63 |
| NMI | 0.95 | 0.90 | 0.87 |

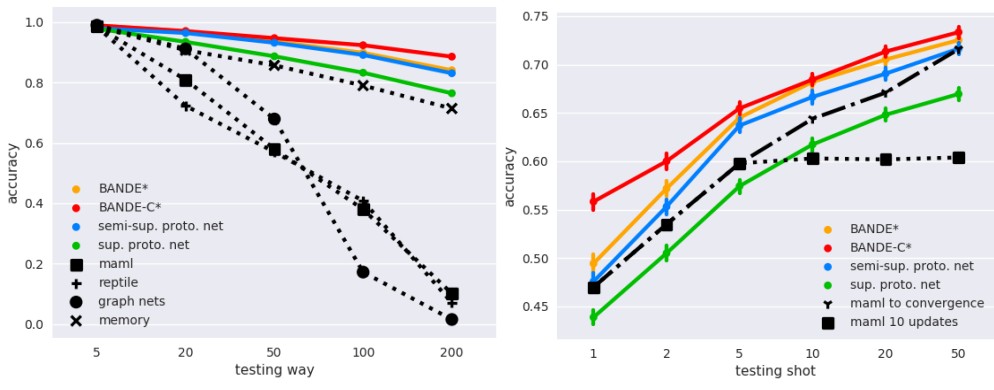

(a) way generalization on Omniglot      (b) shot generalization on mini-ImageNet

Figure 2: Our new variadic setting for any-shot, any-way generalization. Models are meta-trained with 5-way 1-shot episodes. Omniglot is tested with 1-shot episodes across 5–200 classes per episode, while mini-ImageNet is tested with 5-way episodes across 1–50 examples per class. Baselines (black) are trained on $100\%$ of the labeled data. Prototypical methods (color) are semi-supervised with $40\%$ of the labeled data (our methods starred). For way (a), nonparametric methods significantly outperform parametric methods, and BANDE performs best. For shot (b), nonparametric and gradient methods perform similarly but gradient methods require more computation.

## 4.2 ANY-SHOT, ANY-WAY LEARNING IN THE VARIADIC SETTING

We now move to the any-shot, any-way setting that this paper introduces. We closely examine extreme generalization across shot and way between meta-train and meta-test, unlike previous approaches which only examine small shifts (Munkhdalai & Yu, 2017; Snell et al., 2017). Most notably, we show that nonparametric methods such as BANDE can generalize from few-way training to many-way testing, while parametric methods fail to transfer effectively. We further show that BANDE, a *nonparametric* method, performs on par with *fully parametric* methods in the domain of supervised learning; the first demonstration of a meta-learning method evaluated in the many-shot domain without pre-training. These two results cement the suitability of nonparametric meta-learning methods over parametric methods for the variadic setting.

**Semi-supervised protocol** We train and test BANDE and other prototypical methods on *semi-supervised* data to include the number of labeled and unlabeled examples in the scope of the variadic setting. We follow (Ren et al., 2018), taking only $40\%$ of the data as labeled *for both the support and query* while the rest of the data is included, but as unlabeled examples. The unlabeled data is then incorporated into episodes as (1) within support examples that allow for semi-supervised refinement of the support classes or (2) *distractors* which lie in the complement of the support classes. Semi-supervised episodes augment the fully supervised $n$-way, $k$-shot support with 5 unlabeled examples for each of the $n$ classes and include 5 more distractor classes with 5 unlabeled instances each. The query still contains only support classes.

**Variable Shot and Way** We first look at generalization by moderately adjusting the shot and way in evaluation from their fixed settings during meta-learning. For variable way, we consider Omniglot, because it has many classes. For variable shot, we consider mini-ImageNet, because it has more examples per class. In both cases, we train on 5-way, 1-shot episodes, and test generalization by varying the number of classes and number of examples during meta-testing.

We consider four strong fully-supervised baselines trained on $100\%$ of the data (black lines), as well as prototypical baselines trained on $40\%$ of the data (colored). We compare to three parametric methods, MAML (Finn et al., 2017), Reptile (Nichol & Schulman, 2018), and few-shot graph networks (Garcia & Bruna, 2018), as well as the nonparametric memory-based model of Kaiser et al. (2017). Modifications to these approaches for test-way generalization are discussed in Section A.3.

As seen in Figure 2 (a), the parametric meta-learning approaches fail to meaningfully generalize to higher way than they were trained for. BANDE is the least sensitive to higher way meta-testing, although the margin between BANDE and semi-supervised prototypical networks in this regime is small compared to the difference with parametric methods.

For shot generalization, we compare to MAML's accuracy after 10 updates vs. accuracy at convergence. We note that MAML is not able to make effective use of more data unless it is allowed to take

proportionately larger numbers of updates, while our method improves with more data without taking gradients at test time. Even at convergence, MAML lags BANDE's performance, suggesting that a nonparametric approach is still superior to parametric meta-learning.

**Extreme Generalization to Many-Way** We demonstrate that BANDE can learn a full 1692-way classifier for Omniglot from only episodic optimization of 5-way 1-shot tasks. Episodes are composed identically to the few-shot semi-supervised setting with unlabeled examples and distractor classes. Accuracies for our method and a supervised learning baseline are shown in Figure 3.

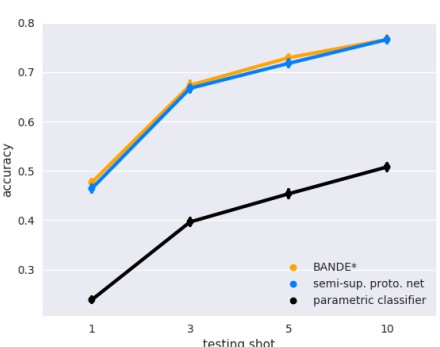

For inference, we run $k$ examples from each test class through our learned embedding network, and then assign the unseen examples the label of the closest prototype. The baseline shares the same training set and architecture, substituting a linear output layer for prototypes by optimizing the softmax cross-entropy loss. We take the last feature layer as the embedding for prototypical inference.

Figure 3: 1692-way classification of all Omniglot test classes from 5-way 1-shot meta-learning. Non-parametric methods can learn this many-way task from few-way meta-training with higher accuracy than a full-way parametric classifier.

Fine-tuning on the test support proved less accurate, as did k nearest neighbours inference.

This result is an example of episodic optimization yielding strong results for many-way classification, motivating the possibility of learning large-scale models cumulatively from small-scale tasks, instead of restricting attention to the adaptation of large-scale models to small-scale, few-shot settings.

**Scaling to Many-Shot** We examine the effectiveness of BANDE in the conventional supervised learning regime. To the best of our knowledge this is the first evaluation of meta-training across the spectrum from few-shot to many-shot. Our base architecture is the Wide ResNet 28-10 of Zagoruyko & Komodakis (2016), which has shown state-of-the-art results on CIFAR-10/100, and has been additionally used as a base architecture for strong low-shot performance on mini-ImageNet (Qiao et al., 2018). We optimize BANDE by meta-training on episodes consisting of 10-way (CIFAR-10) 2-shot and 20-way (CIFAR-100) tasks for computational considerations.

With no knowledge of the total number of classes or number of examples per class, and without pre-training or fine-tuning, we achieve accuracies that rival a well-tuned supervised learning baseline. On CIFAR-10 we achieve $94.4\%$ accuracy compared to the $95.1\%$ accuracy of supervised learning. On CIFAR-100 we achieve $75.6\%$ accuracywhich is $> 90\%$ of the $81.2\%$ accuracy of supervised learning. When evaluating both the BANDE and supervised learning embeddings as prototypes the accuracies are equal, suggesting that both approaches learn equally good representations, and differ only in the prototypical vs. parametric form of the classifier.

### 4.3 FEW-SHOT CLASSIFICATION BENCHMARKS

We evaluate BANDE on the standard few-shot classification benchmarks of Omniglot and mini-ImageNet in the fully-supervised and semi-supervised regimes.

BANDE learns to recover uni-modal clustering as a special case, matching or out-performing prototypical networks when the classes are uni-modal, as seen in Table 4. In this setting, we evaluate BANDE in the standard episodic protocol of few-shot learning. In this protocol, shot and way are fixed and classes are balanced within an episode.

The results reported in Table 4 are for models trained and tested with $n$-way episodes. This is to equalize comparison across methods. Snell et al. (2017) train at higher-way than testing and report a boost in accuracy. We find that this boost is illusory, and explained away by controlling for the number of gradients per update. We show this by experiment through the use of gradient accumulation in Section A.2 of the appendix. (For completeness, we confirmed that our implementation of prototypical networks reproduces reported results at higher way.)

Table 4: Fully-supervised few-shot accuracy using 100% of the labeled data. BANDE performs equal to or better than prototypical networks (Snell et al., 2017). Although BANDE is more general, it can still recover uni-modal clustering as a special case.

| | Omniglot | | | | mini-ImageNet | |
| | 5-way | | 20-way | | 5-way | |
| Method | 1-shot | 5-shot | 1-shot | 5-shot | 1-shot | 5-shot |
|---|---|---|---|---|---|---|
| BANDE (ours) | 98.4±0.1 | 99.5±0.1 | 95.1±0.1 | 98.6±0.1 | 48.9±0.7 | **68.3±0.6** |
| Snell et al. (2017) | 98.3±0.2 | 99.6±0.1 | 94.9±0.2 | 98.6±0.1 | 46.4±0.78 | 67.0±0.7 |
| Finn et al. (2017) | 98.7±0.4 | 99.9±0.3 | 95.8±0.3 | 98.9±0.2 | 48.7±1.84 | 63.1±0.92 |
| Garcia & Bruna (2018) | **99.2** | 99.7 | **97.4** | 99 | **50.3** | 66.41 |
| Kaiser et al. (2017) | 98.4 | 99.6 | 95 | 98.6 | - | - |

Table 5: Semi-supervised few-shot accuracy on 40% of the labeled data with 5 unlabeled examples per class and 5 distractor classes. The distractor classes are drawn from the complement of the support classes and are only included unlabeled. BANDE achieves equal or better accuracy than semi-supervised prototypical networks (Ren et al., 2018).

| | Omniglot | | | | mini-ImageNet | |
| | 5-way | | 20-way | | 5-way | |
| Method | 1-shot | 5-shot | 1-shot | 5-shot | 1-shot | 5-shot |
|---|---|---|---|---|---|---|
| BANDE (ours) | **98.9 ± 0.1** | **99.4 ± 0.1** | **97.0 ± 0.1** | **98.4 ± 0.1** | **49.2 ± 0.7** | **66.1 ± 0.7** |
| Ren et al. (2018) | 98.0 ± 0.1 | **99.2 ± 0.1** | 96.4 ± 0.1 | 98.1 ± 0.1 | **48.6 ± 0.6** | 63.8 ± 0.8 |
| Snell et al. (2017) | 97.8 ±0.1 | **99.2 ± 0.1** | 93.4 ± 0.1 | 98.0 ± 0.1 | 43.9 ± 1.0 | 63.6 ± 0.5 |

In the semi-supervised setting we follow (Ren et al., 2018), using the set-up outlined in the second paragraph of section 4.2. Our results for this setting are reported in Table 5.

Through multi-modality, the clustering of the labeled classes and distractors is decided by the data with a single rule. In particular this helps with the distractor distribution, which is in fact more diffuse and multi-modal by comprising several different classes. Our only specialization to this setting is to have more uncertain distractor clusters by higher cluster variances to compensate for this diffuseness.

## 5 CONCLUSION

We framed the variadic regime to shine a light on learning representations that bridge small-scale and large-scale learning and strive toward the any-shot/any-way adaptability of human perception. As a step toward addressing this full span, we introduced BANDE, a multi-modal extension of prototypical networks, that is capable of generalizing across variable amounts of labeled and unlabeled data. Our results have shown BANDE is state-of-the-art in the few-shot regime and scales from few-way, few-shot meta-learning to many-way, many-shot deployment for both sparse and plentiful supervision. Our experiments demonstrate that multi-modality is key for improved semi-supervised and unsupervised clustering. There is much work to be done to improve variadic generalization, and to connect to life-long learning over non-stationary tasks.

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

## A   APPENDIX

### A.1   IMPLEMENTATION DETAILS

For all few-shot experiments, we use the same base architecture as prototypical networks for the embedding network. It is composed of four convolutional blocks consisting of a 64-filter 3 x 3 convolution, a batch normalization layer, a ReLU nonlinearity, and a 2 x 2 max-pooling layer per block. This results in a 64-dimensional embedding vector for omniglot, and a 1600 dimensional embedding vector for mini-imagenet. Our models were trained via SGD with RMSProp (Tieleman & Hinton, 2012) with an $\alpha$ parameter of 0.9. For Omniglot, the initial learning rate was set to 1e-3, and cut by a factor of two every 2000 iterations, starting at 4000 iterations. We additionally use gradient accumulation and accumulate gradients over eight episodes before making an update when performing 5-way training for Omniglot. For mini-ImageNet, the initial learning rate was set to 1e-3, and further halved every 20000 iterations, starting at 40000 iterations.

For the supervised experiments, we use a wide residual network (Zagoruyko & Komodakis, 2016) with depth 28 and widening factor 10, with a dropout value of 0.3. We were not able to perfectly recover published results with our reimplementation, but the numbers are within $1\%$ of their published values.

### A.2   CONTROLLING FOR THE NUMBER OF GRADIENTS TAKEN DURING OPTIMIZATION

Consider the gradient of the loss: it has the dimensions of shot × way because every example has a derivative with respect to every class. In this way, by default, the episode size determines the number of gradients in an update. Quantitatively, 20-way episodes accumulate 16 times as many gradients as 5-way episodes. By sampling 16 5-way episodes and accumulating the gradients to make an update, we achieve significantly better results, matching the results obtained with 20-way episodes within statistical significance. Note that agreement across conditions may not be perfectly exact because subtle adjustments to hyperparameters might be necessary.

Table 6: Results on Omniglot for different gradient accumulations. Bolded results are not significantly different from each other.

| Shot | Batch-way | Episode-way | 5-way | | 20-way | |
|---|---|---|---|---|---|---|
| | | | 1-shot | 5-shot | 1-shot | 5-shot |
| 1 | 20 | 20 | **98.5** | **99.6** | **95.0** | **98.8** |
| 1 | 20 | 5 | **98.3** | **99.5** | **94.8** | **98.6** |
| 1 | 5 | 5 | 97.7 | **99.4** | 92.1 | 98.0 |
| 5 | 20 | 20 | **97.8** | **99.6** | **93.2** | **98.6** |
| 5 | 20 | 5 | **97.9** | **99.6** | **92.9** | **98.5** |
| 5 | 5 | 5 | 96.8 | **99.4** | 89.8 | 97.7 |

### A.3   EXTENDING COMPARED MODELS TO VARIADIC REGIME

The models we compare to were not designed with variadic generalization in mind, and as a result we attempt to make as fair a comparison as possible by extending them as needed. We describe our approaches below.

**Semi-supervised prototypical networks**   In the paper first introducing this semi-supervised setting (Ren et al., 2018), the authors show how to use a distractor cluster centered at 0 to capture samples not belonging to any examples from the support. They additionally introduce length scales $r_c$. In equation 6 from their paper, they use a normalization constant $A(r_c)$ defined as $0.5\log(2\pi) + \log(r)$. However, this is an unscaled normalization constant, and assumes the dimensionality of the embedding space to be 1. The corrected normalization constant is $A(r_c) = d(\log(r_c) + 0.5\log(2\pi))$ where $d$ is the dimensionality of the embedding. We compare to their method with this corrected normalization constant, but note that it has only a small effect. For space, we did not compare to all methods from their paper, and chose this one as it performed well across their experiments, and because it was most amenable to the clustering experiments we were interested in performing.

**MAML** We used Finn's publicly available github repository (Finn et al., 2017). We trained an initial MAML architecture on the 5-way 1-shot task, using the suggested hyperparameters, for 40,000 iterations. We then removed the classification layer, froze the remaining weights of the network (for optimization across episodes, not for gradient descent within an episode), and retrained the top layer for the testing $n$-way classification task, using the MAML objective again, for 5000 iterations. We tried two hyperparameter settings for the re-training: the hyperparameters for the 5-way 1-shot setting, and the hyperparameters for the 20-way 1-shot setting. We found that re-training with the 20-way 1-shot hyperparameters gave us better performance. While we attempted to also scale these hyperparameters appropriately for even higher way testing, this was not more successful than using the 20-way 1-shot hyperparameters. We then reported the accuracy after 10 update steps on the test data.

We also tried simply randomly initializing the top-layer weights, and allowing MAML to take more update steps to see if it could learn the top layer online. These results were worse than those obtained after the fine-tuning procedure.

**Reptile** We used the publicly available github repository from OpenAI. We used transductive training for 100,000 iterations on the 5-way 1-shot task, using the suggested hyperparameters. We then removed the classification layer, froze the remaining weights of the network, and retrained the top layer for the testing $n$-way classification task, using the Reptile training procedure. As in MAML, we tried setting hyperparameters during re-training to be similar to 5-way 1-shot, and 20-way 1-shot, but did not notice significant differences. Using random initializations for the top-layer weights, and then applying "fast weight" updates at test time also worked reasonably well.

**Graph Neural Networks** Modifying the Graph Neural Network architecture to be applicable for test-way generalization was more difficult, since the approach assumes that labels are represented as a one-hot encoding, and concatenated with node features before being fed to the metric network. At training, we padded the one-hot labels to allow for 200 possible classes. At test time, these could then be filled in without needing to completely retrain the metric network. We additionally fine-tuned the classification layer of the metric network. We were unable to achieve greater than chance performance for the 200-way task. We expect that this is because the metric network learns to ignore the padded input dimensions during training. One possible fix would be to randomize the labels during training to fall in the full (0, 200) range, but we leave this to future work. Scaling this approach up to full-way classification is impossible with this encoding of the labels, as the computational memory requirements are substantial.

A.4   SOFT-SOFT CLUSTERING BY APPROXIMATING THE CHINESE RESTAURANT PROCESS

Here we discuss an alternative to BANDE which follows Gibbs sampling in an infinite mixture model more closely, in that it incorporates variances of clusters throughout, instead of only during reassignment as in BANDE. This fully soft variant has a probabilistic interpretation through the Chinese Restaurant Process (CRP) of Aldous (1985), but in our experiments it achieves lower accuracy than BANDE. For a certain setting of its parameters we can reinterpret it as an infinite mixture model extension of (Ren et al., 2018), which did not include this theoretical perspective.

The generative model of the CRP consists of sampling assignments $z_1, ..., z_J$ which could take on cluster values $c = 1, ..., C$ from the CRP prior with hyperparameter $\alpha$, which controls the concentration of clusters, and number of cluster members $N_c$. Cluster parameters $\mu_c, \sigma_c$ are sampled from a base distribution $H(\theta_0; \mu_0, \sigma_0)$, and instances $x_j$ are then sampled from the associated Gaussian distribution $\mathcal{N}(\mu_{z_j}, \sigma_{z_j})$. $\theta_0$ and $\theta$ consist of the parameters to be estimated, which in this case are the mean $\mu$ and variance $\sigma$ of the Gaussian distributions.

The CRP generative model is defined as

$$p(z_{J+1} = c | z_{1:J}, \alpha) = \frac{N_c}{N + \alpha} \text{ for } c \in \{1 \dots C\} \text{ and } p(z_{J+1} = C + 1 | z_{1:J}, \alpha) = \frac{\alpha}{N + \alpha} \quad (1)$$

for assignments $z$ of examples $x$ to clusters $c$, cluster counts $N_c$, and parameter $\alpha$ to control assignments to new clusters. $N$ is the total number of examples observed so far.

One popular sampling procedure for parameter estimation is Gibbs sampling (Neal, 2000). In Gibbs sampling, we draw from a conditional distribution on the cluster assignments until convergence. The conditional draws are: $p(z_{J+1} = c | z_{1:J}, \alpha) \propto \begin{cases} N_{c,-j} \int P(x_j|\theta) dH_{-j,c}(\theta) \text{ for } c \leq C \\ \alpha \int P(x_j|\theta) dH_0(\theta) \text{ for } c = C + 1 \end{cases}$

For the case of a spherical Gaussian likelihood, let us define $\mathcal{N}_c = \mathcal{N}(x_i; \mu_c, \sigma)$ as the likelihood of assigning $x_i$ to cluster $c$ and $\mathcal{N}_0 = \mathcal{N}(x_i; \mu_0, \sigma + \sigma_0)$ as the likelihood of assigning $x_i$ to a new cluster drawn from the base distribution (Gaussian with mean $\mu_0$ and $\sigma_0$) . We can then write:

$$p(z_i = c | \mu) = \frac{N_{k,-n} \mathcal{N}_c}{\alpha \mathcal{N}_0 + \sum_{j=1}^{C} N_{j,-n} \mathcal{N}_j} \tag{2}$$

$$p(z_i = C + 1 | \mu) = \frac{\alpha \mathcal{N}_0}{\alpha \mathcal{N}_0 + \sum_{j=1}^{C} N_{j,-n} \mathcal{N}_j} \tag{3}$$

$$p(\sigma_c | z) = \frac{\sigma \sigma_0}{\sigma + \sigma_0 N_c} \tag{4}$$

$$p(\mu_c | z) = \mathcal{N}\left(\mu_c; \frac{\sigma \mu_0 + \sigma_0 \sum_{i, z_i = c} x_i}{\sigma_c + \sigma_0 N_c}, \frac{\sigma \sigma_0}{\sigma + \sigma_0 N_c}\right) \tag{5}$$

---

**Algorithm 2** Soft-soft clustering: multi-modal clustering with cluster variances for labeled and unlabeled data by approximating the Chinese Restaurant Process (CRP). $n_s$ is the number of labeled classes (way). $q(i, c)$ is $\log p(i, c)$, the joint probability of cluster $C$ and assignment $i$. $\mathcal{N}(x; \mu, \sigma)$ is the Gaussian density. $\alpha$ is the concentration hyperparameter of the CRP. $\epsilon$ is the threshold hyperparameter for creating a new cluster.

---

initialize $\{\mu_1, \ldots, \mu_{n_s}\}$ ▷ Initialize a cluster for each labeled class by taking class-wise means
initialize $\{\sigma_1, \ldots, \sigma_{n_s}\}$ ▷ Initialize cluster variances based on equation 4.
initialize $\{z_1, \ldots, z_I\}$ ▷ Initialize cluster assignments for labeled data points. All unlabeled cluster assignments start at 0.
$C = n_s$ ▷ Initialize number of clusters $C$
▷ Begin clustering pass
**for** each example $i$ **do**
    **for** each cluster $c \in \{1, ..., C\}$ **do**
        $N_c \leftarrow \sum_i z_{i,c}$
        $\sigma_c \leftarrow \frac{\sigma \sigma_0}{\sigma + \sigma_0 N_c}$
        $\mu_c \leftarrow \frac{\sigma \mu_0 + \sigma_0 \sum_i z_{i,c} h_\phi(x_i)}{\sigma_c + \sigma_0 N_c}$
        estimate $q_{i,c} \propto \log(N_{c,-i}) + \log(\mathcal{N}(x_i; \mu_c, \sigma_c))$ based on equation 2
    **end for**
    estimate $q_{i,C+1} \propto \log(\alpha) + \log(\mathcal{N}_0(x_i; \mu_0, \sigma_0))$ based on equation 3
    $z_{i,c} \leftarrow \text{softmax}(q_{i,1}, ..., q_{i,C+1})$
    **if** $z_{i,C+1} > \epsilon$ **then**
        $C \leftarrow C + 1$
    **end if**
**end for**

---

Determining the assignment for a query sample is performed after clustering using the updated means and cluster counts.

We connect our fully soft clustering variant to prior work on semi-supervised prototypical networks (Ren et al., 2018) to give it a new probabilistic perspective. Their method clusters labeled examples into a cluster per class by class-wise means, defines a "distractor" cluster for unrelated unlabeled examples, and then refines the labeled clusters by soft k-means. Their distractor cluster is fixed to have a mean of zero and variance of 100. If we set $\mu_0 = 0$ and $\sigma_0 = 100$ accordingly, and do not update $\sigma$ parameters, then our fully soft clustering can be seen as the infinite mixture model extension of their method, where the distractor cluster corresponds to a draw from a general base distribution with a CRP prior placed on the cluster assignments.

## A.5 SEMI-SUPERVISED CLUSTERING EVALUATION

We show the importance of multi-modality for discovering unlabeled clusters during meta-testing after semi-supervised meta-learning.

We randomly sampled $n$ classes from Omniglot's test set, and ran one randomly selected example from each of the $n$ classes through our model to obtain a set of $n$ prototypes.

We then presented a new set of examples drawn equally from these $n$ support (known) classes and $n$ out-of-support (unknown) classes, then let each method cluster the examples into either the computed $n$ prototypes or new clusters.

Figure 4 shows the $n+1$ accuracy of correctly classifying the new example as either the correct, known label, or correctly identifying the example as a distractor. Only BANDE achieved higher accuracy than chance for numbers of clusters greater than 5, suggesting that a multi-modal distribution for unlabeled clusters is paramount for the algorithm's clustering performance. The number of clusters created for the unlabeled examples closely tracked the correct number of unlabeled clusters, with an average relative error in the number of created clusters of 1.87 across the range from 5 - 200.

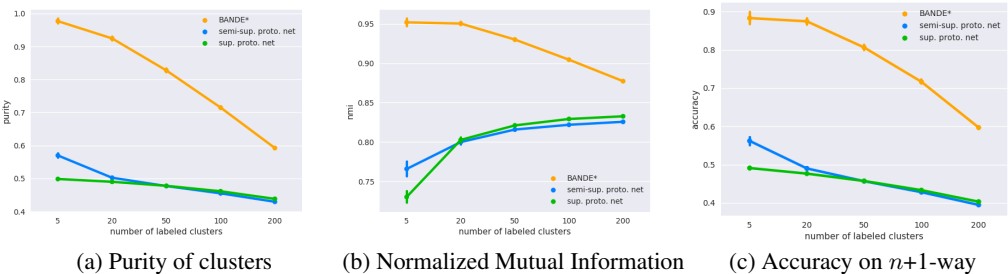

(a) Purity of clusters    (b) Normalized Mutual Information    (c) Accuracy on $n+1$-way

Figure 4: Cluster discovery metrics for Omniglot. Trained on 5-way 1-shot episodes.

