# OpenReview forum: "Variadic Learning by Bayesian Nonparametric Deep Embedding"
_ICLR.cc/2019/Conference_

### Official Review · AnonReviewer1 · 2018-10-26
**A work lacking clarity**

**Rating:** 4
**Confidence:** 4

**Review:**

This work proposes a learning method based on deep subspace clustering. The method is formulated by identifying a deep data embedding, where clustering is performed in the latent space by a revised version of k-means, inspired by the work [1]. In this way, the proposed method can adapt to account for uni-modal distributions. The authors propose some variations of the framework based on soft cluster assignments, and on cumulative learning of the cluster means.
The method is tested on several scenarios and datasets, showing promising results in prediction accuracy.

The idea presented in this work is reasonable and rather intuitive. However, the paper presentation is often unnecessarily convoluted, and fails in clarifying the key points about the proposed methodology. The paper makes often use of abstract terms and jargon, which sensibly reduce the manuscript clarity and readability. For this reason, in my opinion, it is very difficult to appreciate the contribution of this work, from both methodological and applicative point of view.

Related to this latter point, the use of the term “Bayesian nonparametric” is inappropriate. It is completely unclear in which sense the proposed framework is Bayesian, as it doesn’t present any element related to parameters inference, uncertainty estimation, … Even the fact that the method uses an algorithm illustrated in [1] doesn’t justifies this terminology, as the clustering procedure used here only corresponds to the limit case of a Dirichlet Process Gibbs Sampler when the covariance parameters goes to zero. Moreover, the original procedure requires the iteration until convergence, while it is here applied with a single pass only. The procedure is also known to be sensitive to the order by which the data is provided, and this point is not addressed in this work.

Finally, the novelty of the proposed contribution is questionable. To my understanding, it may consist in the use of embedding methods based on the approach provided in [1]. However, for the reasons illustrated above, this is not clear. There is also a substantial amount of literature on deep subspace embeddings that proposes very similar methodologies to the one of this paper (e.g. [2-5]).  For this reason, the paper would largely benefit from further clarifications and comparison with respect to these methods.





[1] Kulis and Jordan,  Revisiting k-means: New Algorithms via Bayesian Nonparametrics, ICML 2012

[2] Xie, Junyuan, Ross Girshick, and Ali Farhadi. "Unsupervised deep embedding for clustering analysis." International conference on machine learning. 2016.
[3] Ji, Pan, et al. "Deep subspace clustering networks." Advances in Neural Information Processing Systems. 2017.
[4] Jiang, Zhuxi, et al. "Variational deep embedding: An unsupervised and generative approach to clustering." IJCAI 2017
[5] Kodirov, Elyor, Tao Xiang, and Shaogang Gong. "Semantic autoencoder for zero-shot learning. CVPR 2017.

---

> ### Author Response · Authors · 2018-11-13
> **Relation/Contrast to Deep Subspace Embedding, Novelty, and Breadth of Results**
>
> > proposes a learning method based on deep subspace clustering
> > substantial amount of literature on deep subspace embeddings that proposes very similar methodologies to the one of this paper (e.g. [2-5])
>
> We thank the reviewer for bringing up deep subspace embedding. While our work and these are generally related by metric learning, they are quite separate in approach and purpose. Ours is a meta-learning approach for multi-modal representation (that is, having an adaptive number of centroids per class) of labeled and unlabeled data, it is optimized for classification tasks, and it is evaluated by generalization to new data and tasks. The cited [2-5] address unsupervised clustering, have fixed numbers of clusters, and are evaluated by clustering metrics on the same data they are optimized on.
>
> Most significantly, these works *do not consider generalization*: the clustering methods are optimized on the data that is to be clustered and do not experiment on held-out tasks/classes as in meta-learning settings like ours. Only [5] can incorporate labeled data, and in their experiments they train and test on the same classes, without generalization, on a tiny synthetic dataset and the Oxford flowers dataset of 17 classes and <1000 images.
>
> [2, 3, 4, 5] learn and evaluate unsupervised and zero-shot clustering models on the same train/test data with the same classes without generalization experiments. [2] cannot incorporate labeled data, requires pre-training, and shows results on the toy datasets of MNIST and STL-10. [3] cannot incorporate labeled data and is only evaluated on the simple face and object datasets Yale B, ORL, and COIL. [4] addresses generative modeling and unsupervised clustering for problems, not few-shot learning and classification, and its experiments are restricted to small-scale datasets with 10 or fewer clusters. [5] focuses on zero-shot learning with a linear auto-encoder on off-the-shelf features, and its "supervised clustering" section has only a 3-class synthetic dataset and a 17-class dataset of flower images where the clustering is optimized for the same 17 flower species it is evaluated on.
>
> > novelty of the proposed contribution is questionable
>
> Here is a brief summary of our key, novel contributions:
>
> technical novelty: our method is capable of adaptive, multi-modal clustering unlike the fixed, uni-modal clustering of Ren et al. and Snell et al. by our reconciliation of DP-means from Kulis et al. with end-to-end learning (section 3.2).
>
> empirical novelty: we propose and thoroughly investigate our "variadic" setting of any-shot/any-way generalization (section 4.2), find that several popular methods degrade in this setting (MAML, Reptile, few-shot graph nets), show that it is possible to learn a large-scale classifier (1692-way character recognition) from small-scale episodic optimization (5-way 1-shot tasks), show that episodic optimization of a prototypical method rivals the accuracy from large-scale SGD optimization of a strong fully-parametric baseline optimized by SGD on CIFAR-10/100, and evaluate few-shot learning of alphabets instead of characters to examine accuracy on more complex data distributions.
>
> theoretical novelty: We shed further light on prototypical network methods with the lens of probabilistic interpretation. We derive an approximate interpretation of Ren et al. (Appendix A4), which lacked theoretical justification, and explain the direct interpretation of the hard variant of our own method (Section 3.4).
>
> > method is tested on several scenarios and datasets, showing promising results in prediction accuracy
>
> We thank the reviewer for commenting on our breadth of evaluation and promising results. To reinforce this point, we note that our experiments cover several problem statements: few-shot fully-supervised/semi-supervised classification (Section 4.1, Tables 1 & 2), our proposed variadic setting of any-shot/any-way generalization (Section 4.2), purely unsupervised clustering (Section 4.3, table 3) and transfer learning from super-class training to sub-class recognition (Section 4.3, table 4). We approach each of these problems by meta-learning through episodic optimization of classification tasks, and these experiments focus on generalization to new tasks (of held-out classes, different settings of shot and way, or discovery of sub-classes from super-class training).

---

> > ### Author Response · Authors · 2018-11-13
> > **Bayesian Nonparametric Name, Terminology, and Clustering Details**
> >
> > > use of the term “Bayesian nonparametric” is inappropriate
> >
> > The DP-means clustering method of Kulis et al., which our work adapts to end-to-end optimization for metric learning, is derived through bayesian nonparametric infinite mixture modeling in the limit of zero variance. The existence of the method, and others that share this mathematical framework (Broderick et al. 2013, Roychowdhury et al. 2013, Wang & Zhu 2015), are due to bayesian nonparametrics and identify as such in their titles and text. Not acknowledging this connection could obscure the origin and properties of the method. Does the reviewer have an alternate term in mind?
> >
> > > paper makes often use of abstract terms and jargon
> >
> > Could the reviewer please be more precise on this point? We have made our best effort to follow the standard terminology for meta-learning and few-shot learning (Vinyals et al. Finn et al., Snell et al, Ren et al.), but would appreciate knowing specifically where this is confusing, so that it can be more clear for a broader audience.
> >
> > > procedure is also known to be sensitive to the order by which the data is provided, and this point is not addressed in this work.
> >
> > While it is true that the clustering is dependent on the order of the data, we simply have not found the method to be sensitive to this in practice, although we can include this result in the revision. We note that this dependence is likewise mentioned in Kulis et al. 2012 but they make no mention of it impacting the quality of their results.
> >
> > > proposed method can adapt to account for uni-modal distributions
> >
> > Our method critically allows for *multi-modality* in the data distribution for both labeled and unlabeled data, adaptively choosing the number of clusters, unlike the prior work by Snell et al. and Ren et al. that assume fixed numbers of clusters, as do [2, 3, 4, 5] cited in the review. This is explained in Section 3.2 and shown to be crucial for diverse classes like alphabets in Section 4.3 Table 4.

---

> > > ### Comment · AnonReviewer1 · 2018-11-25
> > > **reply**
> > >
> > > I thank the authors for their clarification. However, I am still not very convinced about the use of the terminology made in this work.
> > >
> > > The proposed scheme builds upon an algorithm which was obtained as the zero variance limit of a Bayesian mixture model.
> > > This does not justify the term Bayesian non-paramteric for the proposed method. As stated in my first review, the framework does not present any property of the Bayesian methodology such as the possibility of inference over parameters, uncertainty quantification, or model comparison. Even the very standard k-means clustering, or a linear model, can be seen as the limit of Bayesian counterparts, but it would be awkward, and not justified, to present a work using these tools as Bayesian.
> > >
> > > I stil think that there is a substantial problem of novelty and clarity. As also noted by reviewer 3, the difference with respect to the state of the art resides in the use of algorithm 1 for adaptively adding new centres if needed. While being of interest, this part deserves further clarifications on the many critical aspects: stability, dependence on the parameter \lambda, uniqueness of the solution. In the current version of the manuscript these aspects are lightly mentioned and not discussed. The fact that the authors “ have not found the method to be sensitive to this in practice” does not represent a strong motivation in favour of the method. Moreover, as already mentioned in my previous review, the proposed use of the method is different from the original formulation of [Kulis 2012], as it is not iterated (“In this clustering scheme a single pass is sufficient …”). This raises further concerns about stability and robustness of the proposed procedure.
> > >
> > > Finally, I apologise for the previous use of the term “uni-modal distribution”. I acknowledge that the proposed method is explicitly built to account for multi-modal ones, and I made a mistake while typing my previous review.

---

> > > > ### Author Response · Authors · 2018-11-27
> > > > **Alternative terminology, novelty, and clustering quality**
> > > >
> > > > > the framework does not present any property of the Bayesian methodology such as the possibility of inference over parameters, uncertainty quantification, or model comparison
> > > > > Even the very standard k-means clustering, or a linear model, can be seen as the limit of Bayesian counterparts, but it would be awkward, and not justified
> > > >
> > > > We thank the reviewer for articulating this definition of a bayesian method. As we have explained in our first response, unlike k-means, the dp-means algorithm of Kulis et al. 2012 was derived via bayesian nonparametrics and relies on this mathematical framework for inferring the number of clusters, and so we reference this to make the origin and properties of the clustering clear. We welcome the reviewer to suggest an alternative to "bayesian nonparametric" (as we did before), but to be more concrete we would like to ask if "infinite mixture modeling" would be more apt from the reviewer's perspective?
> > > >
> > > > > I stil think that there is a substantial problem of novelty and clarity
> > > > > the difference with respect to the state of the art resides in the use of algorithm 1 for adaptively adding new centres if needed
> > > >
> > > > The technical novelty of our work is in sec. 3 and algorithm 1: we make use of dp-means for inferring the number of clusters, define and experiment on three multi-modal clustering variants, develop a method to choose the cluster distance threshold lambda episodically, and mask assignments and the loss to handle both labeled and unlabeled data.
> > > >
> > > > In our first response, we also highlighted our empirical novelty in exploring any-shot/any-way generalization in our proposed variadic setting and theoretical novelty in deriving a theoretical connection to semi-supervised prototypical networks. Could the reviewer please let us know if they are aware of existing work with these experiments and theory?
> > > >
> > > > > critical aspects: stability, dependence on the parameter \lambda, uniqueness of the solution
> > > > > proposed use of the method is different from the original formulation of [Kulis 2012], as it is not iterated
> > > > > that the authors "have not found the method to be sensitive to this [the order of data] in practice" does not represent a strong motivation
> > > >
> > > > We thank the reviewer for their consideration of clustering quality. We agree this is important, and so we summarize where it is addressed in our work.
> > > >
> > > > - stability and iterations: The clustering converges and multiple iterations maintain the quality of the results for classification (tables 2, 4, and 5 for example) and unsupervised clustering (table 3). While we noted in the text that one iteration was sufficient to achieve our state-of-the-art results, we will edit to explain that multiple iterations are stable in the camera ready version of the paper.
> > > > - dependence on lambda: Our method includes a procedure for choosing lambda that we make use of throughout our experiments (please see section 3, last paragraph). The quality of our results supports this procedure.
> > > > - robustness and order: Noting the lack of sensitivity to the order of the data was a direct response to the reviewer's concern that different orderings might affect the results. Our empirical results show this is not a weakness.
> > > >
> > > > We thank the reviewer for these points, which can be further highlighted in a final revision.
> > > >
> > > > Last, we would like to note that we have posted a revision incorporating feedback from the reviews, and request to know if the reviewer finds it to be more clear.

---

### Official Review · AnonReviewer3 · 2018-11-02
**Novelty is unclear**

**Rating:** 4
**Confidence:** 2

**Review:**

The paper proposes a meta-learning method that utilizes unlabeled examples along with labeled examples. The technique proposed is very similar to the one by (Ren et al. 2018), only differing in the choice of a different clustering algorithm (Kulis and Jordan, 2012) instead of soft k-means as used by Ren et al.

I feel the contrast to Ren et al, is not provided to the degree it should be. The Appendix paragraph A4 is not sufficient in terms of explaining why this method is conceptually different or significantly better than the related approach. It is hard for me to certify the merits of their work, including explaining the experimental results.

I also do not understand the significance of "multi-model clustering" in this context. Also, by their definition of "variadic", how is this more variadic than Ren et al. or Snell et al.?

---

> ### Author Response · Authors · 2018-11-13
> **Contrast with Ren et al., Significance of Multi-modal (Many-to-One) Clustering, and Variadic Setting**
>
> We thank the reviewer for raising three key points of our work: (1) clustering algorithm choices and our difference with Ren et al., (2) our technical contribution of extending prototypical methods to multi-modal representation for handling more complicated data distributions, and (3) our empirical contribution of proposing and thoroughly investigating the variadic setting of any-shot/any-way generalization.
>
> > the contrast to Ren et al, is not provided to the degree it should be
> > only differing in the choice of a different clustering algorithm
>
> The difference in choice of clustering is crucial:
> - our method is capable of adaptive, multi-modal clustering unlike the fixed, uni-modal clustering of Ren et al. and Snell et al. This gives an improvement of +3 points accuracy on the standard few-shot benchmark of 5-way, 5-shot mini-ImageNet classification (Table 2), extends prototypical nets to problems without any labeled data (see next bullet point), and for more diverse classes like alphabets our accuracy is ~25 points higher.
> - our method handles labeled data by the same clustering rule unlike the heuristics of Ren et al. for unlabeled data, making inference in our method possible for zero labeled examples (of any kind, including meta-data as in zero-shot learning) whereas Ren et al. and Snell et al. are undefined in this setting. Section 4.3 shows high quality clustering without labels (Table 3), and 10-25 point improvements on prior work for learning more diverse classes like alphabets instead of single characters (Table 4), underlining the importance of multiple modes.
> - We shed further light on the choice of clustering with the lens of probabilistic interpretation: we derive an approximate interpretation of Ren et al. (Appendix A4), which lacked theoretical justification, while explaining the direct interpretation of the hard variant of our own method (Section 3.4).
>
> > significance of "multi-model clustering"
>
> Multi-modality is a key and distinguishing property of our method that is necessary for the quality of our results. Please refer to figure 1 for a schematic of the difference among Snell et. al, Ren et al., and BANDE (ours): note that having multiple modes lets BANDE more accurately cluster the labeled and unlabeled data alike. Among these methods, only BANDE can adjust its capacity to model simple, compact classes with a single mode while simultaneously modeling diverse, complicated classes with multiple modes. We achieve higher accuracy than Ren et al. for semi-supervised few-shot learning (Table 2). Furthermore, Table 4 in particular highlights the needs for multi-modal representation: a full alphabet is not uni-modal in the learned embedding, unlike a single character, and here we show major (10-25) point gains over the prototypical nets of Snell et al. and Ren et al. that assume each class has a uni-modal data distribution.
>
> > by their definition of "variadic", how is this more variadic than Ren et al. or Snell et al.?
>
> Snell et al., Ren et al., and our method do indeed generalize better across shot and way as we show (Figure 2). Our first contribution is in evaluating this generalization at all in our novel experiments: we cover extreme way at 1692 Omniglot classes (Figure 3), extreme shot at zero labeled examples for clustering (Table 3) and at scaling episodic optimization to the supervised learning regime of 50k labeled examples on CIFAR-10 and CIFAR-100. Existing work was restricted to the few-shot settings of Section 4.1 with training/testing on the same way and shot.
>
> BANDE (ours) is more variadic than Ren et al. and Snell et al. in 1. handling the case of purely unlabeled data and 2. handling more diverse data with complicated class distributions such as alphabet classes instead of character classes (section 4.3). We forecast that meta-learning, as it scales to more diverse data distributions, will encounter more tasks like our alphabet recognition experiments in the variety and even hierarchy of classes, where our adaptive, multi-modal clustering helps significantly (Table 4). While we expect further progress to improve on Ren et al., Snell et al., and our own method, the main point here is to encourage this kind of shot/way generalization to reconcile the distant poles of small-scale and large-scale learning.

---

> > ### Comment · AnonReviewer2 · 2018-11-25
> > **Proposed method's multi-modality seems distinct from previous work to me**
> >
> > R3, any revised thoughts on novelty based on this careful feedback from the authors?
> >
> > Seems to me that the difference between previous methods and the current approach is given in Fig. 1.... previous methods assume each class has a single center in the learned feature space (e.g. the left panel in Fig. 1), while the proposed approach allows each class to have multiple centers if needed (the far right panel). This multi-modality makes the proposed method more flexible.

---

### Official Review · AnonReviewer2 · 2018-11-02
**Hard to read and relies on unjustified, shifting assumptions**

**Rating:** 5
**Confidence:** 4

**Review:**

Update after Author Rebuttal
--------------
After reading the rebuttal, I'm pleased that the authors have made significant revisions, but I still think more work is needed. The "hard/soft" hybrid approach still lacks justification and perhaps wasn't compared to a soft/soft approach in a fair and fully-correct way (see detailed reply to authors). I also appreciate the efforts on revising clarity, but still find many clarity issues in the newest version that make the method hard to understand let alone reproduce. I thus stand by my rating of "borderline rejection" and urge the authors to prepare significant revisions for a future venue that avoid hybrids of hard/soft probabilities without justification.

(Original review text below. Detailed replies to authors are in posts below their responses).

Review Summary
--------------
While the focus on variadic learning is interesting, I think the present version of the paper needs far more presentational polish as well as algorithmic improvements before it is ready for ICLR. I think there is the potential for some neat ideas here and I hope the authors prepare stronger versions in the future. However, the current version is unfortunately not comprehensible or reproducible.

Paper Summary
-------------

The paper investigates developing an effective ML method for the "variadic" regime, where the method might be required to perform learning from few or many examples (shots) and few or many classes (ways). The term "variadic" comes from use in computer science for functions that can a flexible number of arguments. There may also be unlabeled data available in the few shot case, creating semi-supervised learning opportunities.

The specific method proposed is called BANDE: Bayesian Nonparametric Deep Embedding. The idea is that each data point's feature vector x_i is transformed into an embedding vector h(x_i) using a neural network, and then clustering occurs in the embedding space via a single-pass of the DP-means algorithm (Kulis & Jordan 2012). Each cluster is assumed to correspond to one "class" in the eventual classification problem, though each class might have multiple clusters (and thus be multi-modal).

Learning occurs in an episodic manner. After each episode (single-pass of DP-means), each point in a query set is embedded to its feature vector, then fed into each cluster's Gaussian likelihoods to produce a normalized cluster-assignment-probability vector that sums to one. This vector is then fed into a cross-entropy loss, where the true class's nearest cluster (largest probability value) is taken to be the true cluster. This loss is used to perform gradient updates of the embedding neural network.

There is also a "cumulative" version of the method called BANDE-C. This version keeps track of cluster means from previous episodes and allows new episodes to be initialized with these.

Experiments examine the proposed approach across image categorization tasks on Omniglot, mini-ImageNet, and CIFAR datasets.


Strengths
---------
* I like that many clusters are used for each true class label, which is better than rigid one-to-one assumptions.


Limitations
-----------
* Can only be used for classification, not regression
* The DP-means procedure does not account for the cluster-specific variance information that is used at other steps of the algorithm


Significance and Originality
----------------------------
To me, the method appears original. Any method that could really succeed across various variadic settings would be significant.



Presentation Concerns
---------------------

I have serious concerns about the presentation quality of this paper. Each section needs careful reorganization as well as rewording.

## P1: Algo. 1 contains numerous omissions that make it as written not correct.

* the number of clusters count variable "n" is not updated anywhere. As writting this algo can only update one extra cluster beyond the original n.
* the variable "c" is unbound in the else clause. You need a line that clarifies that c = argmin_{c in 1 ... n} d_ic

Would be careful about saying that "a single pass is sufficient"... you have *chosen* to do only one pass. When doing k-means, we could also make this choice. Certainly the DP-means objective could keep improving with multiple passes.

## P2: Many figures and tables lack appropriate captions/labels

Table 1: What metric is reported? Accuracy percentage? Not obvious from title/caption. Should also make very clear here how much labeled data was used.

Table 2: What metric is reported? Accuracy percentage? Not obvious from title/caption. Should also make how many labeled and unlabeled examples were used easier to find.

## P3: Descriptions of episodic learning and overall algorithm clarity

Readers unfamiliar with episodic learning are not helped with the limited coverage provided here in 3.1 and 3.2. When exactly is the "support" set used and the "query" set used? How do unlabeled points get used (both support and query appear fully labeled)? What is n? What is k? What is T? Why are some points in Q denoted with apostrophes but not others? Providing a more formal step-by-step description (perhaps with pseudocode) will be crucial.

In Sec. 3.2, the paragraph that starts with "The loss is defined" is very hard to read and parse. I suggest adding math to formally define the loss with equations. What parameters are being optimized? Which ones are fixed?

Additionally, in Sec. 3.2: "computed in the same way as standard prototypical networks"... what is the procedure exactly? If your method relies on a procedure, you should specify it in this paper and not make readers guess or lookup a procedure elsewhere.


## P4: Many steps of the algorithm are not detailed

The paper claims to set \lambda using a technique from another paper, but does not summarize this technique. This makes things nearly impossible to reproduce. Please add such details in the appendix.

Major Technical Concerns
------------------------

## Alg. 1 concerns: Requires two (not one) passes and mixes hard and soft assingments and different variance assumptions awkwardly

The BANDE algorithm (Alg. 1) has some unjustified properties. Hard assignment decisions which assume vanishing variances are used to find a closest cluster, but then later soft assignments with non-zero variances are used. This is a bit heuristic and lacks justification... why not use soft assignment throughout? The DP means procedure is derived from a specific objective function that assumes hard assignment. Seems weird to use it for convenience and then discard instead of coming up with the small fix that would make soft assignment consistent throughout.

Furthermore, The authors claim it is a one pass algorithm, but in fact as written in Alg. 1 it seems to require two passes: the first pass keeps an original set of cluster centers fixed and then creates new centers whenever an example's distance to the closest center exceeds \lambda. But then, the *soft* assignment step that updates "z" requires again the distance from each point to all centers be computed, which requires another pass (since some new clusters may exist which did not when the point was first visited). While the new soft values will be close to zero, they will not be *exactly* zero, and thus they matter.

## Unclear if/how cluster-specific variance parameters learned

From the text on top of page 4, it seems that the paper assumes that there exist cluster-specific variances \sigma_c. However, these are not mentioned elsewhere, only a general (not cluster-specific) label variance \sigma and fixed unlabeled variance sigma_u are used.

## Experiments lack comparison to internal baselines

The paper doesn't evaluate sensitivity to key fixed hyperparameters (e.g. \alpha, \lambda) or compare variants of their approach (with and without soft clustering step, with and without multimodality via DP-means). It is difficult to tell which design choices of the method are most crucial.

---

> ### Author Response · Authors · 2018-11-13
> **Clarity, Reproducibility, and Details to Resolve Technical Concerns**
>
> We thank the reviewer for their detailed feedback, in particular the attention to the technical aspects of the clustering steps and probabilistic interpretations in our work, and the comments on clarity and accessibility for audiences less familiar with meta-learning and few-shot learning. We agree with the reviewer on the importance of multi-modal clustering as "better than rigid, one-to-one assumptions" of prior work, which we show by experiment on alphabet recognition in section 4.3 and improved semi-supervised few-shot classification in section 4.1. We likewise agree that methods that "really succeed across various variadic settings would be significant" which is why we propose it in this work and investigate it by experiment in section 4.2.
>
> Regarding concerns of clarity and reproducibility, we are incorporating the feedback of the reviews into a revision to be posted during the rebuttal period and will release code after decision (omitted here only to preserve anonymity). Our comprehensive code release will cover our model, experimental evaluation and training settings, all few-shot baselines (including prototypical networks, semi-supervised prototypical networks, and our variadic extensions of MAML and few-shot graph networks), and datasets. This will help safeguard reproducibility for future work and serve as a reference implementation of the variadic setting.
>
> We now clarify our method and experiments to address the reviewer's technical concerns.
>
> hard/soft assignments and probabilistic interpretation: We thank the reviewer for their theoretical precision. We are in full agreement, and wish to point out that we identify and experiment with fully hard (sec. 3.4) and fully soft variants of our method (appendix A4) for this reason of probabilistic justification. We choose the hard-soft hybrid for our main results, as mentioned in the paper, because it is marginally more accurate in experiments. We appreciate the feedback on this point, and are revising the text to make these variants more clear.
>
> number of passes/clustering steps: We will clarify our language to use the term “clustering iteration” instead of passes/clustering steps. In the fully hard model, an iteration corresponds to the assignment of all labeled and unlabeled points to clusters, and then an update of the means of all clusters. In the fully soft model, an iteration corresponds to computing soft assignments for all points, and then updating the means. In the hard-soft hybrid, we use the “hard” step to compute a set of cluster means, and then perform a "soft" clustering step in order to update these cluster means.
>
> cluster-specific variances: \sigma and \sigma_u are learned and are shared across all labeled and all unlabeled clusters respectively. \sigma_c was a typo for \sigma as it is the variance of class clusters. The only exception to learning these variances, as noted, is Section 4.3 where they are fixed.
>
> internal baselines/ablations: We agree with the reviewer on this list of ablations/internal baselines, so much so that we have already experimented with them in the development of the method: the selected multi-modal clustering with hard-soft assignment was best. For exposition we chose to focus on the hard-soft variant as our method and compare to competing works like Ren et al. and Finn et al., but for completeness we will include these ablation experiments in our revision to the text (to be posted during the rebuttal period).
>
> “our method can only be used for classification, and not regression”: While true, this weakness holds for prior prototypical methods too by Snell et al. and Ren et al. so our work is no more and no less limited in this regard.

---

> > ### Author Response · Authors · 2018-11-13
> > **Incorporation of Presentation Feedback (Thanks!)**
> >
> > We now turn to the reviewer's thorough feedback on presentation.
> >
> > P1: numbers of iterations. In principle, BANDE can be iterated multiple times, as in DP-means. However, in our experiments we found accuracy does not improve with more iterations. We will modify the text to make this more clear. We will also correct the algorithm description to appropriately update n and c with the required two lines (thank you for catching this).
> >
> > P2: Tables 1 and 2 captions and details. The metric is indeed accuracy percentage, as is standard for these benchmarks, which we are clarifying in our revision (to be posted during the rebuttal period). We appreciate that the semi-supervised setting of Ren et al. has a number of details, which is why we placed the paragraph on semi-supervised episode composition under table 2, and we will incorporate more of this text into the caption to make it easier to find.
> >
> > P3: episodic learning. We would like to thank the reviewer for commenting on the clarity of our work for readers who do not specialize in meta-learning and few-shot learning. We tried to follow the standard summary in this field (see Ren et al., Finn et al.). While fuller tutorial coverage of few-shot learning would be the most clear, we are constrained by the page limit when explaining the existing settings and our contributions of multi-modality and any-shot/any-way generalization in new settings. We are clarifying few-shot details in captions and the main text in the revision.
> >
> > P4: setting λ. The technique for setting λ is our own, which we summarize at the end of 3.2: "We estimate ρ as the variance in the labeled cluster means within an episode, while α is treated as a hyperparameter." The algebraic expression of λ in terms of ρ, α is what we borrow from Kulis et al., and we are rewording this for clarity in the revision.
> >
> > "computed in the same way as standard prototypical networks" (section 3.2). This is explained in the last paragraph of 3.1, so we remove it here to avoid redundancy and potential confusion.

---

> > > ### Comment · AnonReviewer2 · 2018-11-25
> > > **Thanks for your revisions! Quality is improving, but there still some issues that make me reluctant to accept**
> > >
> > > P1: I appreciate the expanded Alg. 1. Definitely an improvement, but there are still some significant issues.
> > >
> > > * The first line uses "C", but "C" hasn't been defined. I think you mean the total number of labeled classes in the dataset "n_s".
> > > * You should clarify that p(x | mu, sigma) is the Gaussian PDF (e.g. a specific function that evaluates to a probability density)
> > > * The distance computation of d_ic as written asks if y_i == c. But I think you really want to test if y_i == \ell_c (the label of class c). Otherwise you'll never be able to reuse new clusters you create, since those clusters will have c > n_s and thus y_i == c will always be false.
> > > * In the final cross entropy expression, the variable "c" is unbound in the right hand term. You want it to be defined by the max, but as written it is a separate variable.
> > >
> > > There are also key sentences in the paper that are misleading/unclear. For example, the sentence " Unlike DP-means, we include cluster variances" is a bit odd.... the paper does NOT use any variances for the DP-means part of Alg. 1. However, it does use some variance parameters for later stages. So they haven't changed the DP-means algorithm to include variances, they just use variances in a post-processing step

---

> > > > ### Author Response · Authors · 2018-11-27
> > > > **Incorporation of latest feedback (thanks!)**
> > > >
> > > > Thank you again to the reviewer for their attention to detail. We have incorporated these comments into the algorithm in the revision.
> > > >
> > > > >There are also key sentences in the paper that are misleading/unclear.
> > > > >For example, the sentence " Unlike DP-means, we include cluster variances" is a bit odd....
> > > >
> > > > We have clarified the language in the methods section with respect to DP-means and our contributions. For example, we changed the above sentence to “While we use DP-means for cluster creation, we include cluster variances for reassignment.” Is the latest revision clearer?
> > > >
> > > > We would like to ask the reviewer, given their significant feedback on the theoretical components of the work, if they could comment on the practical contributions of the work for meta-learning. We are thankful for the improvements in the clarity and accessibility of the work due to the reviewer's comments, and we would appreciate further comments on the contributions of the work with respect to prototypical networks and meta-learning methods more generally.

---

> > ### Comment · AnonReviewer2 · 2018-11-25
> > **Hard/soft approach still lacks justification ....**
> >
> > Thanks for clarification on many details. I'm glad you have planned a code release. I hope the revised manuscript also includes enough details that readers don't have to go look at code for every detail.
> >
> > I'm still not satisfied by the proposed hard-soft hybrid. I suppose it may be "marginally more accurate in experiments", though I only see one test on one dataset, where soft-soft accuracy is 98.4 and hard-soft accuracy is 99.0, a difference that seems too small to justify an approach that is more complicated and not very well founded compared to one that has a much better interpretation (and thus more confidence it will work on other datasets).
> >
> > It's also not clear that a fair, correct experiment was done between "soft-soft", "hard-hard", and "hard-soft". Alg. 1 (hard-soft) can reassign both labeled and unlabeled data to new clusters. However, Alg. 2 (the hard-hard method) in A.4 only operates on unlabeled examples. (Also, the value of \sigma_0 is unclear). No formal algorithm is given at all for the soft-soft method.  Note also that the condition for creating a new cluster in Alg. 1 is that the distance to the closest cluster exceeds some threshold. However, in the written hard-hard algorithm, the condition is different: distance to some \mu_0 which is fixed to 0.0. The better approach would be to use the MARGINAL likelihood of x_i being assigned to a new cluster, as in the Gibbs sampler for DP mixtures (see Eq. 3.7 of Radford Neal's "Sampling methods for Dirichlet Process Mixture Models" (http://www.stat.columbia.edu/npbayes/papers/neal_sampling.pdf).
> >
> > Given all these concerns, it's unclear if there's really a fair comparison here.

---

> > > ### Author Response · Authors · 2018-11-27
> > > **Hard/soft clarification, empirical justification, and verification of experiment correctness**
> > >
> > > We thank the reviewer for their continued attention to the theoretical aspects of the work.
> > >
> > > > still not satisfied by the proposed hard-soft hybrid
> > > > an approach that is more complicated and not very well founded compared to one that has a much better interpretation
> > >
> > > We sympathize with the reviewer’s hope for the empirical dominance of the theoretically pure variants of our method, but in practice we have found that their accuracies are worse than existing results, as well as our hard-soft hybrid. We have incorporated the reviewer’s feedback into our exposition of the hard-soft hybrid and the theoretical ramifications of our choices in the revision. We hope our theoretical coverage and empirical investigation of these variants sets the stage for future work to further reconcile theory and practice.
> > >
> > > > though I only see one test on one dataset
> > > > (and thus more confidence it will work on other datasets)
> > >
> > > We have also experimented on mini-ImageNet to cover both standard few-shot learning benchmarks. In the 5-way 1-shot setting with 5 unlabeled examples per class the results are: BANDE is 49.2%, Ren et al. is 48.6%, soft-soft is 47.1%, and hard-hard is 43%. In further experiments at different shot and way the methods keep this ordering. We included only the most common Omniglot setting in the revision for brevity, but can include a full appendix in the camera-ready version. As an alternative, we could compare all three variants throughout our experimental section. Would the reviewer find that more clear? In the existing text we focused on hard-soft for simplicity of description and comparison with prior work, but could revise this for the camera-ready.
> > >
> > > > not clear that a fair, correct experiment was done between "soft-soft", "hard-hard", and "hard-soft"
> > >
> > > Thank you for raising this important point: we assure the reviewer that the experiments comparing the three variants are correct and fair w.r.t. operation over labeled and unlabeled data and new cluster creation. We have clarified this in the revision of appendix A.4. All three variants have the same labeled and unlabeled scope for reassignment. The difference in clustering condition for algorithm 2 derives from the approximation to the Chinese Restaurant Process as a draw from the base distribution and as such is part of the algorithm. The extended A.4 in the revision better delineates the soft-soft variant, which operates on both labeled and unlabeled examples, and its connection to an infinite mixture extension of Ren et al., which is only valid when iterating over unlabeled examples alone and holding variances constant.
> > >
> > > > better approach would be to use the MARGINAL likelihood of x_i being assigned to a new cluster, as in the Gibbs sampler for DP mixtures
> > >
> > > We have clarified A.4 to better indicate that algorithm 2  is the “soft-soft” method, which follows exactly the suggestion of using the marginal probabilities to create a new cluster as in the Neal citation.
> > >
> > > To review, we locate the hard-soft, hard-hard, and soft-soft clustering variants in the text of the revision for further consideration:
> > >
> > > - hard-soft is our chosen method given by algorithm 1 and explained in sec. 3.
> > > - hard-hard is a variant that does not include cluster variances and differs only in the "UpdateAssignments" step of algorithm 1, as detailed in sec. 3.1.
> > > - soft-soft is a variant that incorporates variances throughout, and because it requires more derivation and explanation it is found in appendix A.4 and its algorithm 2, which are referenced from sec. 3.1.
> > >
> > > All three clustering variants presented in this work are novel approaches for end-to-end, multi-modal clustering that extend the accuracy and scope of prototypical networks. We do not intend or claim this work to have a complete theory for nonparametric meta-learning methods, but instead seek to explain the theoretical aspects of our own contributions and existing work on semi-supervised prototypical networks (Ren et al. 2018) that did not have a theoretical interpretation. We leave further theoretical investigation to future work.

---

### Author Response · Authors · 2018-11-28
**Summary of revision**

Thank you to all reviewers for useful feedback on the submission. We have posted a revision with the following changes:

Method:
Overall, we edited the method section to make the algorithm more clear, give a clearer introduction to meta-learning and episodic optimization, and better delineate our contributions relative to DP-means and prototypical networks.

--Sec 3.1 “Foundations” section was removed and incorporated into the introduction of the section, under the headings “few-shot meta-learning”, “prototypes” and “multi-modal clustering”.
--Sec 3.2 (“multi-modal clustering”) was wrapped into section 3 with the bolded “multi-modal clustering”.
--Sec 3.3 (“cumulative supervision”) was moved to Sec 3.2
--Sec 3.4 was moved to Sec 3.1 (“Probabilistic Interpretations and Alternatives” to “Probabilistic interpretations of hard and soft clustering”)
--Sec 3.5 (Implementation details) was redistributed closer to where it was referred to (as suggested by reviewers).
--Algorithm 1 was significantly expanded, to include detailed loss computation, cluster creation and assignment steps, and clearer definitions for all variables.

Results:
We re-ordered the results section to highlight the importance of multi-modality for super-class classification and unsupervised clustering (section 4.1), followed by our variadic setting with extreme-way and extreme-shot results (section 4.2) and finally confirming SOTA performance for few-shot learning (section 4.3). Captions are expanded throughout to ease standalone interpretation of the tables and figures.

---

### Meta-Review · Area_Chair1 · 2018-12-15
**Good but not good enough**

**Confidence:** 5
**Recommendation:** Reject

**Metareview:**

All reviewers wrote strong and long reviews with good feedback but do not believe the work is currently ready for publication.
I encourage the authors to update and resubmit.